# Protein-protein interactions and metabolite channelling in the plant tricarboxylic acid cycle

Youjun Zhang[1], Katherine F.M. Beard[2], Corné Swart[1], Susan Bergmann[1], Ina Krahnert[1], Zoran Nikoloski[1], Alexander Graf[1], R. George Ratcliffe[2], Lee J. Sweetlove[2], Alisdair R. Fernie[1] & Toshihiro Obata[1]

Protein complexes of sequential metabolic enzymes, often termed metabolons, may permit direct channelling of metabolites between the enzymes, providing increased control over metabolic pathway fluxes. Experimental evidence supporting their existence *in vivo* remains fragmentary. In the present study, we test binary interactions of the proteins constituting the plant tricarboxylic acid (TCA) cycle. We integrate (semi-)quantitative results from affinity purification-mass spectrometry, split-luciferase and yeast-two-hybrid assays to generate a single reliability score for assessing protein–protein interactions. By this approach, we identify 158 interactions including those between catalytic subunits of sequential enzymes and between subunits of enzymes mediating non-adjacent reactions. We reveal channelling of citrate and fumarate in isolated potato mitochondria by isotope dilution experiments. These results provide evidence for a functional TCA cycle metabolon in plants, which we discuss in the context of contemporary understanding of this pathway in other kingdoms.

[1] Max-Planck-Institut für Molekulare Pflanzenphysiologie, Am Mühlenberg 1, 14476 Potsdam-Golm, Germany. [2] Department of Plant Sciences, University of Oxford, Oxford OX1 3RB, UK. Correspondence and requests for materials should be addressed to A.R.F. (email: fernie@mpimp-golm.mpg.de).

In 1985 (ref. 1), Srere proposed the word 'metabolon' for a supramolecular complex of sequential metabolic enzymes. Metabolons tend to be non-covalently bound transient complexes allowing the regulation of a metabolic pathway flux by dynamic association and/or dissociation[2–5]. Metabolons mediate 'substrate channelling' (also known as metabolic channelling), wherein reaction intermediates are isolated from the bulk environment surrounding them. Various metabolic advantages of substrate channelling have been postulated, including the following: local enrichment of metabolite to achieve high reaction rate, isolation of intermediates from competing reactions, protection of unstable intermediates and sequestration of cytotoxic metabolites[5–8]. A number of metabolons have already been proposed to mediate substrate channelling in various organisms; for instance, branched chain amino-acid metabolism in human mitochondria[9], the glycolytic pathways of mammals, yeast and plants[10–12] and a wide variety of specialized metabolic pathways including polyamine[13], isoprenoid[14], alkaloid[15] and phenylpropanoid (for example, lignin, carotenoid, flavonoid, isoflavonoid and cyanogenic glucoside[5,16–21]) synthesis in plants. However, the experimental evidence for metabolons remains relatively scarce, despite the fact that the formation of metabolons has been much discussed as a regulatory mechanism in central metabolism[3,5,22] and many modelling studies assume their operation[23–25]. Strictly speaking, metabolite channelling must be observed, for example, by isotope dilution experiments[26], in order for a metabolon to be demonstrated. On the basis of this requirement there is limited evidence for functional metabolons in plants: only glycolysis[10] and the cyanogenic glucoside biosynthetic pathway[21] are shown to operate in this way. Evidence for many of the other pathways postulated to exist as metabolons in plants is based on co-localization of the constituent enzymes by co-purification or Förster resonance energy transfer-based methodologies[13–16].

Although not studied in detail in plants, the tricarboxylic acid (TCA) cycle was the subject of much of early work of Srere on metabolons. It is a ubiquitous metabolic pathway that serves to produce NADH for oxidative phosphorylation and organic acids for biosynthesis in both photosynthetic and heterotrophic organisms and tissues. However, the role of the cycle is markedly different in illuminated photosynthetic cells, since under such conditions photosynthesis dominates the production of reducing equivalents. Despite considerable cumulative evidence of the molecular mechanisms regulating individual enzymes, the molecular basis enabling pathway regulation of the plant TCA cycle still remains to be elucidated[27]. The recent observation that several steps of the TCA cycle are subject to redox regulation by thioredoxin goes some way to addressing this deficit[28]. However, the potential of the constituent enzymes of the plant TCA cycle to interact is yet to be experimentally addressed. By contrast, the possibility of the TCA cycle operating as a metabolon has been much studied in mammalian and yeast systems[29–37], with the term 'metabolon' even being coined for this pathway[1]. Binary interaction studies of the enzymes have also been provided using the bacterial-two-hybrid assay in *Bacillus subtilis*[38]. Moreover, recent studies have provided structural evidence consistent with substrate channelling[36] and demonstrated that dynamics of protein association can be guided by metabolite concentration gradients in a microfluidic cell[37].

In the current study, we aimed to identify possible association among all 38 mitochondrial proteins mediating the TCA cycle and related reactions using four different methods to test binary protein–protein interactions (Supplementary Fig. 1). Our approach revealed a dense protein–protein interaction network of the plant TCA cycle enzymes including interactions between enzymes mediating not only sequential, but also non-sequential

reactions. Additional isotope-based dilution experiments performed with isolated mitochondria provided evidence for substrate channelling that is mediated by only a subset of the enzymes.

## Results

**Construction of the plant TCA cycle interactome.** Thirty-eight mitochondrial proteins of *Arabidopsis thaliana* including all constitutively expressed TCA cycle enzyme subunits and the related mitochondrial proteins mediating the pyruvate dehydrogenase reaction and bypass reactions (namely the NAD$^+$-dependent malic enzymes (MEs) and NADP$^+$-dependent isocitrate dehydrogenase (ICDH)) were tested for binary protein–protein interactions (Supplementary Fig. 1 and Supplementary Table 1). Two quantitative assays and one semiquantitative protein interaction assay were employed to gain a reliable binary protein interaction map of plant mitochondrial TCA cycle proteins. Affinity purification–mass spectrometry (AP–MS) assays were conducted using the PSB-D *Arabidopsis* cell culture line[39] and a green fluorescent protein (GFP)-tag-based affinity purification procedure. In addition, high-throughput split luciferase (split-LUC) assays in *Arabidopsis* mesophyll protoplasts[40] and semiquantitative yeast-two-hybrid (Y2H) assays were also employed for binary protein interaction analysis. It is important to note that these approaches are based on different principles (affinity purification and split molecular complementation) and were performed in widely different physiological conditions (*Arabidopsis* heterotrophic cell culture, mesophyll protoplast and yeast cells). As such, it can be anticipated that complementary results from all three approaches will identify genuine interacting protein pairs.

For the AP–MS assays, normalized signal intensities were processed with the SAINT software to determine fold change-A (FC-A) scores[41], while the relative luminescence unit was quantified for the split-LUC assays. The strength of protein interactions in the Y2H assay was quantified as the initial date of colony observation at 3, 5 or 8 days following inoculation, with earlier colony detection taken to correspond to a stronger interaction. On the basis of the distribution of the data (Fig. 1a–c), we considered as positive interactions the protein pairs for which the scores were in the top 10% (corresponding values of 3.9 and 3.7 in AP–MS and split-LUC, respectively). For the Y2H assays, protein pairs for which colonies were observed at either 3 or 5 days post inoculation were regarded as positive interactions. The higher score from a reciprocal test-pair (that is, proteinA/proteinB and proteinB/proteinA) was considered as the score for the protein pair by each technique. Using these criteria, 124, 126 and 421 binary protein interactions were positive in AP–MS, split-LUC and Y2H assays, respectively (Fig. 1d). Among these putative interactors, 140 protein pairs were detected by at least two methods, hereafter named multiple approach detection, and considered as interacting (Fig. 1d and Supplementary Table 2). Eleven protein pairs were detected by all three methods comprising the most reliably interacting pairs (Supplementary Table 2). These 11 pairs included six interactions between pyruvate dehydrogenase complex (PDC) components. All of the PDC subunits other than lipoamide dehydrogenase (LPD) 2 were included in at least one of the six interactions. This indicates very robust interactions within the enzyme complex, which is likely related to the stability of PDC activity in plant mitochondrial extracts[42] and as such act as a good proof that the approach is effective at identifying interactions. This group of interactions also included those of different isoenzymes of aconitase (ACO; ACO2/ACO3) and subunits of isocitrate dehydrogenase (IDH; IDH2/IDH5) as well as succinate

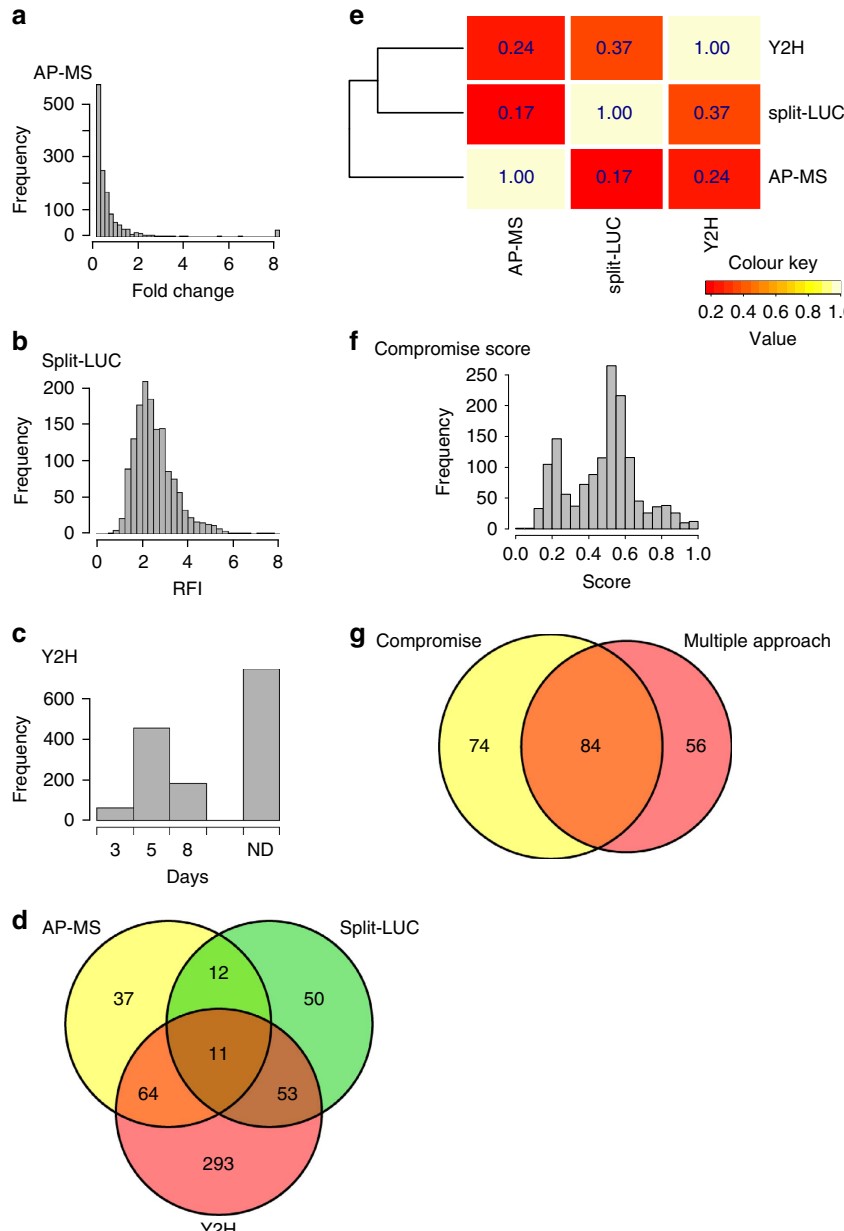

**Figure 1 | Analysis of the data obtained from three independent protein–protein interaction assays.** (**a–c**) Distribution of the scores. Histograms showing distribution of the scores obtained from AP–MS (**a**), split-LUC (**b**) and Y2H (**c**) assays. (**d**) Comparison of the protein pairs detected by each assay. The Venn diagram shows the numbers of the binary protein interactions detected by individual assays. (**e**) Contribution of the results from each technique to the compromise scores. The heatmap shows the eigenvalue decomposition of the cosine matrix generated by the compromise-based method. (**f**) Distribution of the compromise score. (**g**) Comparison of the protein pairs detected by the compromise-based and the multiple approach detection-based methods. The Venn diagram shows the numbers of interactions annotated by single and multiple analyses.

dehydrogenase (SDH; SDH1-1/SDH2-3; Fig. 1d and Supplementary Table 2).

We next integrate the results from the three approaches to generate a single score for each protein–protein interaction that quantifies the reliability of detection. The collection of scores over all binary protein interactions provides a compromise set of interactions among the three methods. For this purpose we applied an approach similar to STATIS[43] to the data matrices gathered by the three methods. The eigenvalue decomposition of the cosine matrix (Fig. 1e), quantifying the distance between each pair of data sets, revealed that AP–MS showed the smallest contribution to the compromise score of interactions followed by LUC and Y2H. These values were in accordance with the block

structure of the compromise matrix, whereby split-LUC and Y2H cluster together (Fig. 1e). The entries of the compromise matrix were bi-modally distributed, with a dominant mode ∼0.6 (Fig. 1f). A truncated set of interactions was obtained by selecting all protein pairs whose compromise score was above this threshold value. Following this approach, the compromise-based method identified a similar number of interactions to those detected in the selection based on the multiple approach detection. We identified a total of 158 protein–protein interactions among the 38 investigated proteins (21.3% of the 741 possible interactions; Supplementary Table 3). Eighty-four interactions were identified by both the compromise-based and multiple approach detection methods, although 74 and 56 of the

interactions were specifically detected by the former and latter analyses, respectively (Fig. 1g). Since the compromise-based analysis includes the quantitative information of each experiment, further analysis and discussion are based solely on the interactions identified by the compromise-based approach. Scores from all individual analytical methods and compromised scores for all tested protein pairs are shown in Supplementary Data 1.

**Catalytic subunits mediating sequential reactions interact.** The entire protein–protein interaction network of TCA cycle enzymes is shown as Fig. 2a. A protein complex likely corresponds to a clique (that is, a subnetwork of mutually connected nodes) in the extracted network. We identified a total of 312 cliques, with the largest clique composed of five proteins, namely, ACO2, ACO3, citrate synthase (CSY) 4, IDH1 and IDH2. In addition, there were 18 cliques each composed of four proteins. Interactions between subunits in known protein complexes, including the PDC (Fig. 2b), the oxoglutarate dehydrogenase complex (ODC; Fig. 2c) and SDH (also known as respiratory electron transport complex II, Fig. 2d) were well captured by our approach. In addition, interactions among catalytic and regulatory subunits of IDH (Fig. 2e) and succinyl-CoA ligase (SCoAL; Fig. 2f) and a previously reported interaction between isoforms of ME[44] were also detected (Fig. 2a). Our analysis

further identified many novel interactions. Arguably, the most important finding, however, was that of interactions between catalytic subunits of enzymes mediating sequential reactions in the TCA cycle. These include the interactions of fumarase (FUM) 1 with malate dehydrogenase (MDH) 2, MDH1 and 2 with CSY4, CSY4 with ACO2 and 3 and ACO3 with IDH6 (Fig. 2g). These interactions were further confirmed by complementary bimolecular fluorescent complementation (BiFC) assays, which enable qualitative but highly sensitive detection of protein–protein interactions with subcellular localization information[45]. For all protein pairs tested, BiFC signals were detected in the mitochondria (Fig. 3a–f). Importantly, no BiFC signals were detected for any of the enzyme pairs, which were not defined as interactors in any of the above approaches (Fig. 3g and Supplementary Fig. 2). These results provide further evidence for the validity of the identified interactions. The interaction between the catalytic subunits potentially reflects metabolon-mediated metabolite channelling. However, interactions between non-catalytic subunits or catalytic subunits of non-sequential reactions may well serve to constrain the enzymes in close proximity. Such interactions include those detected between ACO and IDH (ACO2/IDH1, ACO2/IDH2, ACO3/IDH1 and ACO3/IDH1), IDH and ODC (IDH2/ODC2-1, IDH2/LPD2, IDH5/ODC2-2, IDH6/ODC2-2, IDH6/ODC2-1), ICDH and ODC (ICDH/ODC2-1), ODC and SCoAL (LPD2/SCoALb), SCoAL and SDH (SCoALb/SDH2-2, SCoALb/SDH4), PDC and CSY

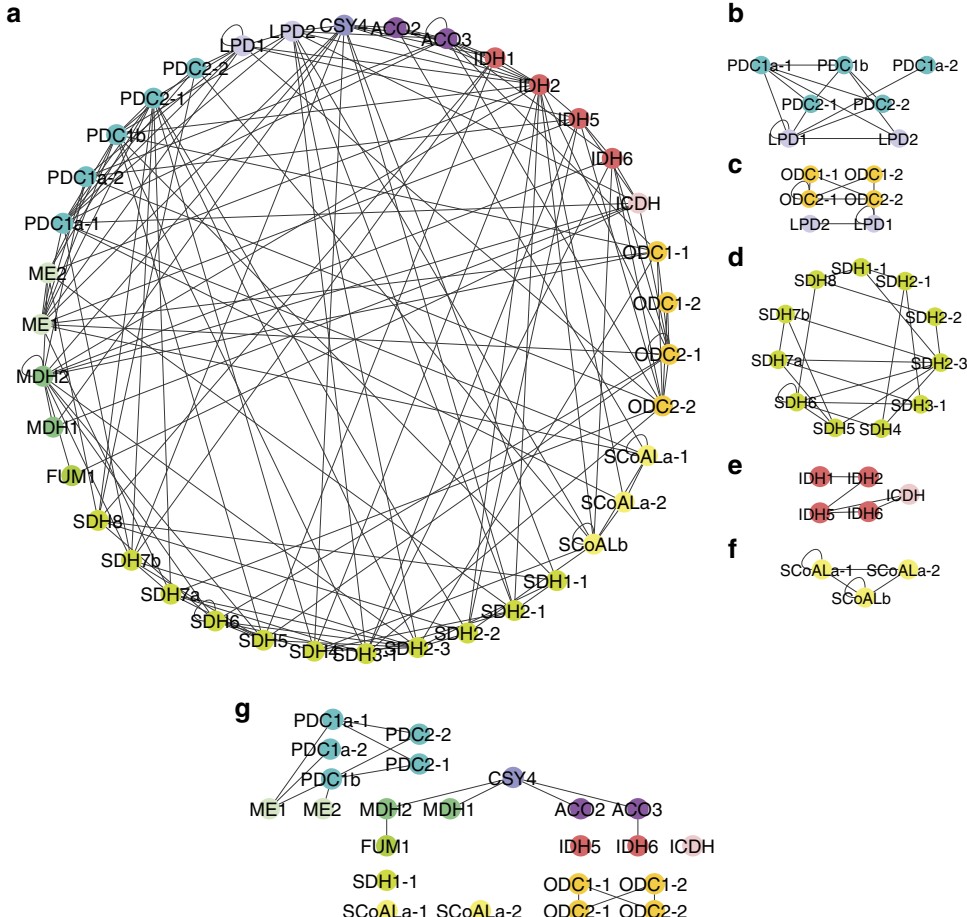

**Figure 2 | Graphical representation of the binary protein–protein interaction network of *Arabidopsis* TCA cycle enzymes.** Node colour represents the enzyme subunits and isoforms. (**a**) Overview of all detected interactions. The proteins are arranged according to the reaction sequence. (**b**–**f**) interactions within enzyme protein complexes of PDC (**b**), ODC (**c**), SDH (**d**), NAD$^+$ dependent-IDH and NADP$^+$ dependent-IDH (ICDH; **e**) and (**f**). (**g**) Interactions between catalytic subunits of enzymes that produce and consume an intermediate metabolite. These interactions potentially mediate metabolite channelling.

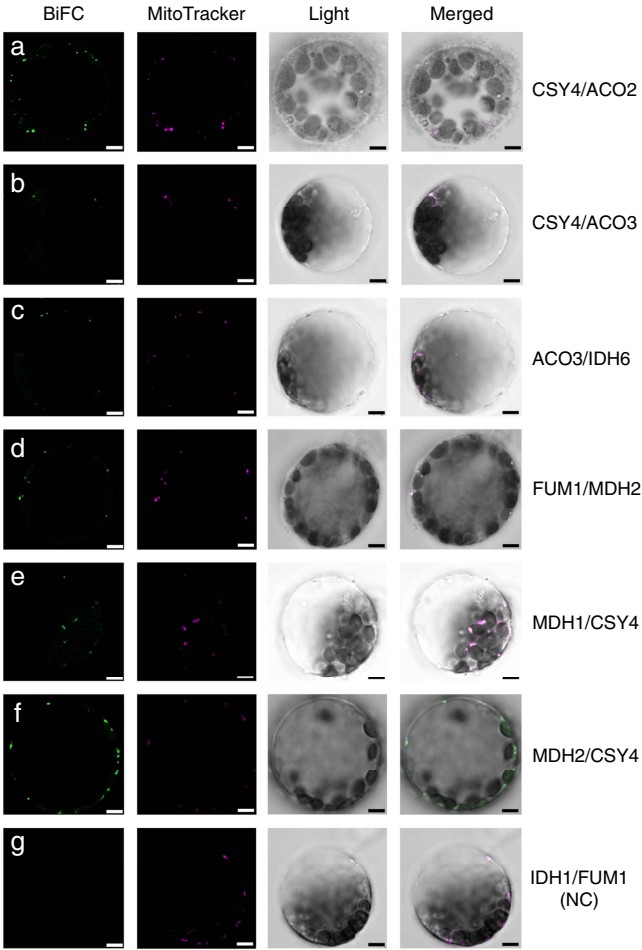

**Figure 3 | Confirmation of selected protein–protein interactions by bimolecular fluorescent complementation assay.** Interactions between catalytic subunits of enzymes mediating sequential reactions of TCA cycle (shown in Fig. 2g) were further tested by BiFC with transient expression of tagged proteins in *Arabidopsis* mesophyll protoplasts. The panels from the left side show the BiFC fluorescence, fluorescence from MitoTracker orange staining, bright field image and the merged image of all of those, respectively. Scale bars, 5 μm. The detail of the constructs can be found in Supplementary Table 6. (**a**) CSY4-SCYCE/ACO2-VYNE; (**b**) CSY4-SCYCE/ACO3-VYNE; (**c**) IDH6-SCYCE/ACO3-VYNE; (**d**) FUM1-SCYCE/MDH2-VYNE; (**e**) CSY4-SCYCE/MDH1-VYNE; (**f**) CSY4-SCYCE/MDH2-VYNE; (**g**) FUM1-SCYCE/IDH1-VYNE. FUM1-SCYCE/IDH1-VYNE is shown as a representative negative control, which was detected by none of the three interaction assays. The images of other negative controls are found in Supplementary Fig. 2.

(LPD2/CSY4) and MDH and IDH (MDH2/IDH2, MDH2/IDH6). Interestingly, our analysis additionally identified interactions between MEs and α-subunits of pyruvate dehydrogenase (PDC1a-1, PDC1a-2 and PDC1b), which can thus be considered as a novel candidate metabolon-mediating pyruvate channelling (Fig. 2g). However, further experiments will be required to conclusively demonstrate both its *in vivo* operation and physiological importance.

**Citrate and fumarate are channelled in isolated mitochondria.** While the above experiments demonstrate protein–protein interaction of the constituent proteins of the TCA cycle including those between the catalytically active subunits of sequential

enzymes of the cycle, they do not directly test the occurrence of metabolite channelling. In order to obtain such information, isotope dilution experiments are necessary[26]. For this purpose mitochondria were isolated from potato tuber tissue, given the ease of isolation from this tissue. The amino-acid sequences of TCA cycle enzymes are quite similar between *Arabidopsis* and potato showing higher than 80% positive match in most subunits (Supplementary Table 4), suggesting similar enzymatic properties in these two species. The isolated mitochondria are incubated with a $^{13}$C-labelled substrate until the label accumulation in the product reached an isotopic steady state (that is, the fractional enrichment of label was constant with time). Then, an unlabelled intermediate of the TCA cycle was added and the 'dilution effect' on the labelling in the product was monitored over time. When the metabolites are being channelled, negligible mixing of pathway intermediates with the environment surrounding the enzyme complex occurs[6] and the unlabelled intermediate has limited access to the enzyme active site. In this scenario, there will be a lack of dilution of $^{13}$C-label in the product. For the purpose of our experiments the TCA cycle was linearized by inhibition of a single reaction to avoid the complication of the results by transfer of label through multiple rounds of the cycle.

The simultaneous application of an inhibitor and a $^{13}$C-labelled TCA cycle intermediate as a substrate leads to continuous accumulation of the labelled product, which is the substrate of the inhibited enzyme. Indeed, succinate and citrate accumulated over the time course when the cycle was inhibited by either malonate or fluorocitrate, respectively (Supplementary Fig. 3). This demonstrates that carboxylic acid metabolism remains active for the duration of the experiment. In addition, ATP synthesis continues in the presence of the inhibitors (albeit at a lower rate as would be expected because of a lower yield of reductant from the linearized pathways) showing that the respiratory chain and ATP synthase also remain functional (Supplementary Fig. 4). The fractional $^{13}$C enrichment in the products increased over time and reached a steady state (Supplementary Fig. 5). Once the isotopic steady state had been reached (35 and 60 min for succinate and citrate, respectively) the deviation in fractional enrichment was within the range of 0.14% and 0.06% for succinate and citrate, respectively (Supplementary Fig. 5). Then, an unlabelled intermediate of interest is added into the system. When it has an access to the catalytic enzyme (that is, the x intermediate is not channelled), the label accumulation in the product will decrease.

$^{13}$C-labelled pyruvate was used as a substrate to test the channelling of citrate and 2-oxoglutarate (2OG). The SDH reaction was inhibited by the competitive inhibitor malonate to linearize the TCA cycle, and the label accumulation in succinate was monitored (Fig. 4a). The steady-state fractional enrichment of succinate was unaltered by the addition of citrate (Fig. 4b), while it decreased following 5 min of incubation with unlabelled 2OG (Fig. 4c). In separate experiments, $^{13}$C-labelled glutamate was fed as substrate and label accumulation in citrate was analysed to test the channelling of succinate, fumarate and malate. The cycle was linearized by inhibition of the ACO reaction by the application of the non-competitive inhibitor fluorocitrate (Fig. 4d). The steady-state fractional enrichment of citrate decreased 20 min after the addition of unlabelled succinate (Fig. 4e), although the addition of fumarate resulted in no reduction of label (Fig. 4f). Addition of unlabelled malate led to a slight reduction of label in citrate following 15 min of incubation (Fig. 4g). The precision with which the labelling can be quantified is more than sufficient to reliably detect changes of the magnitude shown (see Supplementary Fig. 5 for an analysis of analytical precision). These results suggest that citrate and fumarate are channelled, whereas 2OG, succinate and malate are at best only partially channelled in plant mitochondria.

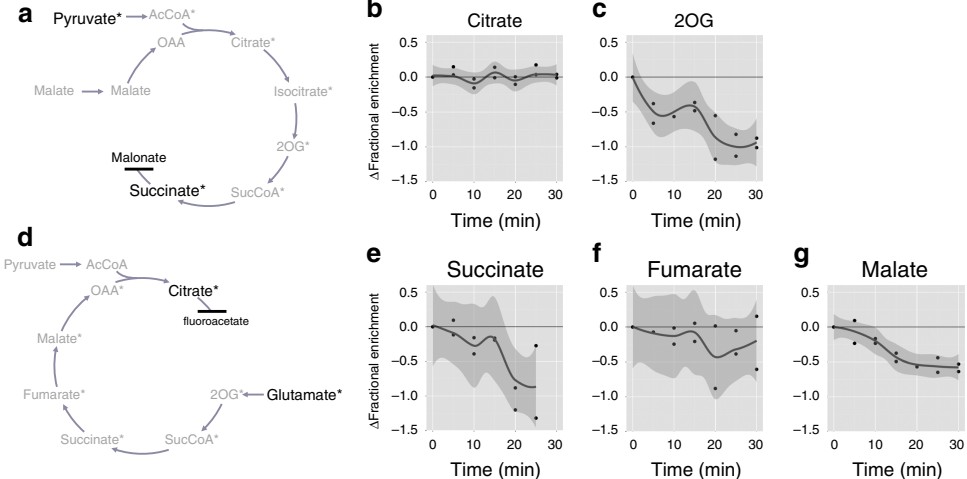

**Figure 4 | Channelling of TCA cycle intermediates in isolated plant mitochondria. (a)** Schematic representation of the isotope dilution experiment to assess the channelling of citrate and 2OG. [13]C-labelled pyruvate was fed to isolated potato mitochondria and the label accumulation in succinate was monitored. The TCA cycle reaction was inhibited by malonate to avoid the complication of multiple turns of the cycle. Non-labelled citrate and 2OG were added into the medium once the fractional enrichment of [13]C-label in succinate had reached steady state. In the case of channelling, the addition of non-labelled intermediates does not affect the subsequent labelling of succinate. **(b)** The result of isotope dilution experiments for citrate. The time course plot shows the changes in fractional [13]C enrichment in succinate following the addition of unlabelled citrate at 0 min. Each point represents the difference in fractional enrichment of label (%) from that at 0 min in each of duplicated samples. The line is the smoothed conditional mean with the shadow showing confidence interval of 0.95. The metabolite is considered not to be channelled when the confidence interval line comes below 0. **(c)** The result of isotope dilution experiments for 2OG. **(d)** Schematic representation of the isotope dilution experiment to assess the channelling of succinate, fumarate and malate. The principle is the same as that for citrate and 2OG but [13]C-glutamate was fed and the label accumulation in citrate was monitored in the presence of fluorocitrate that inhibits further metabolism of citrate. **(e–g)** The results of isotope dilution experiments for succinate **(e)**, fumarate **(f)** and malate **(g)**.

The presence of a branch point at malate (the TCA cycle branch consisting of malate to OAA to citrate and the ME branch consisting of malate to pyruvate to acetyl CoA) complicates the interpretation of the effect of addition of unlabelled malate. However, the effect of the ME branch on isotope dilution in citrate is expected to be minor since unlabelled malate is converted by ME to unlabelled pyruvate, which was added to the system in excess to maintain the pathway flux. Note that other potential complicating branch pathways are either not operating in our system (for example, glutamate dehydrogenase cannot convert 2OG to glutamate because we have not supplied ammonium) or do not change the interpretation of our data. The channelling of isocitrate, succinyl-CoA and oxaloacetate could not be tested because of low stability and/or lack of efficient transport into mitochondria[46].

## Discussion

The advent of proteomics has resulted in a dramatic increase in our understanding of functional protein–protein interactions allowing important advances in various model organisms[47–52]. However, information of the interaction networks of soluble enzymes is scarce even though the microcompartmentation of enzymes is often postulated as an important component of metabolic regulation[3,5,22]. It is probable that this is largely because of the transient nature of this type of protein–protein interaction, which enables dynamic regulation but complicates the detection of protein associations. Here we employed two quantitative methods and one semiquantitative method to reliably capture relatively weak interactions. A well-established affinity purification procedure using *Arabidopsis* cell culture enables a high-throughput AP–MS analysis[39]. Single-step purification using GFP nanobody was employed to facilitate the detection of weak interactions, which can be washed away by tandem affinity purification systems. Low specificity of single-step purification

should be compensated by the combination with two other methodologies. All expression vectors were efficiently constructed from a single Entry vector using Gateway technology enabling the reliable assessment of binary protein–protein interactions of one to dozens of protein pairs. This facilitates a comprehensive analysis of all possible interactions between proteins constituting and associated with the plant TCA cycle. While only 11 interactions were detected by all three methods, a lack of consensus between different methods for identifying protein–protein interactions has previously been documented[52]. It likely reflects the high degree of false-negatives obtained when basing conclusions on a single methodology. It is expected that employing multiple techniques will have a higher chance of detecting weak interactions that are likely to be affected by protein microenvironments. Usually the multiple approach detection of protein–protein interaction relies on qualitative data or uses the quantitative results for setting thresholds within the individual techniques. The compromise-based method we developed here considers the quantitative information from all three technologies for more reliable detection by fine-tuning the thresholds according to the data distribution in each method. Moreover, screening using multiple approaches that require fundamentally different techniques contributes to the elimination of false-positives.

Metabolon formation and metabolite channelling are complex phenomena and require multiple levels of confirmation to prove their occurrence *in vivo*[26]. Historically, metabolon studies initiated from the detection of aggregation of enzymes *in vitro*[31]. This work was followed by *in vitro* enzymatic characterization using purified complexes and structural analysis including modelling approaches[29,30,33,34,53,54]. However, it is very difficult to exclude the possibility of artefacts inducing protein aggregation *in vitro* because of the microenvironment of enzymes, which is significantly different from *in vivo* situation. Recent development of protein–protein

interaction assays provides the possibility to assess metabolon formation under *in vivo* conditions[13,14,17–19], but most of these studies rely on a single approach or at best dual approaches limiting the robustness of the conclusions drawn. The protein interaction takes place in the mitochondria of living *Arabidopsis* cell in three of four methods employed in this study; therefore, our results should reflect more the *in vivo* situation than the previous studies. In addition, the channelling of intermediates between enzymes should be tested by methodologies such as transient-time analysis, isotope dilution/enrichment, competing reaction, enzyme buffering and orientation-conserved label transfer[26]. Especially the isotope dilution experiment employed here and the orientation-conserved label transfer provide a direct proof of metabolite channelling and can be applied *in vivo*. The orientation-conserved label transfer relays on the symmetric structure of fumarate and malate and cannot be applied for assessing substrate channelling of other compounds. This leaves the isotope dilution experiment as the method of choice for testing the entire pathway. However, the application of this technique to *in vivo* studies has been limited to relatively few studies to date[10,21] because of technical difficulties.

In the present study, the combination of interactome analysis, based on multiple experimental approaches, and channelling assays using isolated mitochondria demonstrates the existence of functional metabolons within the plant TCA cycle and affords us an overview of their composition. This is the first systematic study simultaneously assessing both protein–protein interaction and metabolite channelling in the TCA cycle in any organism. It should be noted that our results provide *in vivo* evidence of protein–protein interactions and metabolite channelling, while the knowledge on TCA cycle metabolon in other biological systems is largely dependent on the results of *in vitro* protein aggregation and enzyme kinetics. Nevertheless, the plant TCA cycle metabolon possesses the composition and characteristics previously inferred in bacteria, yeast and mammals. The MDH/CSY/ACO metabolon is the most extensively characterized in the TCA cycle following the first finding of co-precipitation of enzymes from pig heart[55]. Channelling of oxaloacetate in this metabolon was observed in the enzyme complex *in vitro*[56] and the model of a quinary structure of MDH/CSY/ACO metabolon was proposed[54]. The interactions of MDH/CSY and CSY/ACO were recently detected in beef heart mitochondria using a chemical cross-linking-mass spectrometry-based approach and the adjacent protein regions were suggested, on the basis of structural considerations, to support metabolite channelling[36]. We detected interactions between isoforms of MDH, CSY and ACO and citrate channelling in mitochondria, indicating a functional MDH/CSY/ACO metabolon in plants. This part of the TCA cycle metabolon can be considered as the most solid and well-conserved since it has been detected in various organisms across kingdoms[31,38].

In addition to the 'core' part of the TCA cycle metabolon, an interaction of FUM1/MDH2 was detected. It should be noted that the interaction of FUM1 with MDH1, a major mitochondrial isoform of MDH in *Arabidopsis*[57], was not detected. This probably results in a partial channelling of malate, which appeared as slight reduction of label accumulation in the isotope dilution experiment with unlabelled malate. The interaction between FUM and MDH has been reported in rat but it requires chemical cross-linking for detection, suggesting it is rather unstable[32]. The interaction of ACO3/IDH6 suggests further extension of metabolon in plants while the channelling of isocitrate could not be tested because of the inefficient uptake of this compound by plant mitochondria[46]. The association of IDH to the multienzyme complex has been described in bacteria, although it should be noted that the bacterial enzymes are

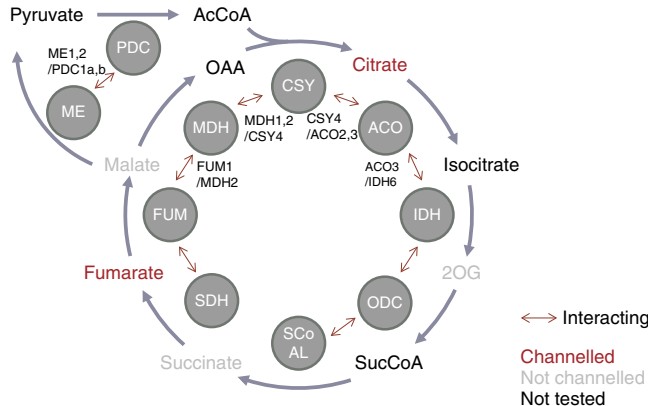

**Figure 5 | Summary of plant TCA cycle metabolon.** The enzymes are depicted as circles and the interactions of any subunits of sequential enzymes are shown as red arrows. The interactions of catalytic subunits that potentially mediate metabolite channelling are described next to the arrows. Metabolites drawn as red, grey and black text are channelled, not channelled and not tested, respectively. PDC, pyruvate dehydrogenase complex; ME, malic enzyme; CSY, citrate synthase; ACO, aconitase; IDH, isocitrate; ODC, oxoglutarate dehydrogenase complex; SCoAL, succinyl-CoA ligase; SDH, succinate dehydrogenase; FUM, fumarase; MDH, malate dehydrogense; AcCoA, acetyl-CoA; 2OG, 2-oxoglutarate; SucCoA, succinyl-CoA.

NADP$^+$-dependent ICDH and different from those in eukaryotes dependent on NAD$^+$ (refs 38,58). Overall, our results strongly indicate the formation of functional TCA cycle metabolon in plant mediating sequential reactions converting fumarate to isocitrate. Corroborating evidence for this is provided by a metabolic flux analysis study, in which the best explanation of $^{13}$C-label redistribution by a heterotrophic *Arabidopsis* cell culture included channelled flux from fumarate or malate to citrate but no channelled flux from 2OG or succinate to citrate[59]. The extension of the metabolon to IDH is novel in eukaryotes, although evidence of channelling is currently lacking. The absence of direct interactions between the catalytic subunits of IDH/ODC, ODC/SCoAL and SCoAL/SDH is in consistent with the results of the isotope dilution experiment showing no channelling of 2OG and succinate. Meanwhile, interactions were detected between IDH and the E2/E3 subunits of ODC as well as the E3 subunit and regulatory subunit of SCoAL (SCoALb). These interactions cannot mediate metabolite channelling but possibly function as a bridge connecting enzyme complexes. This again corresponds to mammalian enzymes for which a multienzyme complex of IDH/ODC/SCoAL has been observed, but no evidence on metabolite channelling within it has been reported[33,34]. While the channelling of fumarate was seen in isolated mitochondria, the interaction of FUM1/SDH1-1 (catalysing the conversion from succinate to fumarate) was not detected. This is similar to the mammalian metabolon in which the physical interaction of SDH/FUM has not been shown despite *in vivo* evidence of fumarate channelling in yeast[35]. This could be because of the requirement of additional *in vivo* components, such as membrane structure and additional protein-anchoring subunits, to form direct interaction of these enzymes.

Channelling was observed in plant mitochondria for citrate and fumarate but not for 2OG, succinate and malate. Although this result does not entirely exclude the possibility of 2OG and succinate channelling, it seems highly unlikely that these metabolites are channelled considering the lack of interaction between the catalytic subunits of the enzymes that produce or consume them (Fig. 5). A lack of channelling of these metabolites

is logical considering that the exchange of these two metabolites with extra-mitochondrial compartments is necessary to support the biosynthesis of a wide range of metabolites including glutamate and to allow the entry of the succinate produced by the GABA shunt into the TCA cycle. On the other hand, the channelling of citrate and fumarate indicates the necessity of dissociation of the metabolon under certain conditions since these metabolites are also known to be secreted from the mitochondria. It will be interesting in future studies to investigate how the composition of the TCA cycle metabolon is dynamically regulated by environmental and developmental inputs including the availability of light and the consequence of such regulation on the respiratory flux[60]. Similarly, experiments have revealed that the proportion of cytosolic isoforms of glycolytic enzymes bound and forming metabolite channels at the outer mitochondrial membrane increases under conditions when respiratory demand is elevated[10].

A number of interactions between subunits of enzymes mediating non-adjacent reactions in the TCA cycle were identified in this study. Such 'crossing' interactions including the interactions of ACO/FUM, ICDH/MDH, ICDH/FUM and SCoAL/MDH have also been detected in B. subtilis[38]. The interaction between ICDH and MDH is promoted in the presence of isocitrate and $NADP^+$, while the activity of ICDH is enhanced by the binding to MDH[61]. Arabidopsis ICDH also interacted with both MDH1 and 2 in this study, suggesting that ICDH activity may also be regulated in a similar manner in plants. It is conceivable that this interaction provides a mechanism for coordination of the levels of the pyridine nucleotides within the mitochondrial matrix, given that the balance between the activities of $NAD^+$-dependent IDH and $NADP^+$-dependent ICDH is considered to play a major role in balancing these[62]. Other such crossing interactions presumably have regulatory roles or act as structural stabilizers of the protein complexes. Metabolons are considered to require structural support from membrane lipids and/or structural proteins that serve as 'nucleation' sites for their formation[16,31,33]. Within our data set IDH1, IDH2 and SCoALb, which do not possess the activities of their corresponding enzymes and are therefore considered as regulatory subunits[63,64], as well as SDH subunits 5–8, which have been identified as plant-specific subunits of the SDH complex but information concerning their function is still lacking[65], likely fulfill such a function. However, further studies will be required in order to fully ascertain the role and relative importance of these proteins.

In the present study, the combination of a protein–protein interaction study with an isotope dilution experiment demonstrated the existence of a plant TCA cycle metabolon. Similar metabolons have been suggested in other systems; however, generally speaking, these rely on a single or at best dual approach and very few present both biophysical evidence of protein interaction and biochemical evidence of metabolite channelling. Although our system relays on the proteins from deferent sources (that is, Arabidopsis and potato), the results should represent the composition of TCA cycle metabolon in plants considering its conserved nature. The unique dynamic properties of plant cellular energy metabolism[60] render it an attractive system in order to evaluate, in future studies, the effects of metabolon organization on metabolic flux control.

## Methods

**cDNA cloning and vector construction.** The mitochondrially localized Arabidopsis proteins involved in the TCA cycle were selected by reference to the literature[27]. Expression of genes encoding these proteins was evaluated in the AtGenExpress 'development' data set[66] in order to select those that were ubiquitously expressed. The proteins catalysing reactions closely related to the

TCA cycle, namely pyruvate dehydrogenase, $NADP^+$-dependent IDH and ME, were also included in the analysis; thus, the list totalled 38 proteins (Supplementary Fig. 1 and Supplementary Table 1). Full-length coding sequences of these proteins were cloned from a cDNA pool generated from 2-week-old A. thaliana Col-0 ecotype plants by PCR-based Gateway BP cloning using the pDONR207 Donor vector (Thermo Fisher Scientific, Waltham, MA). The gene-specific primers used did not include a stop codon to ensure C-terminal fusion of tags (Supplementary Table 5). Expression vectors for AP–MS, split-LUC and Y2H were constructed using the Gateway LR reaction with pK7FWG2 (ref. 67), pDuExAc6/pDuExDc6 (ref. 68) and pGBKCg/pGADCg (ref. 69) Destination vector systems, respectively. The Destination vectors for BiFC were constructed by replacing the sequence encoding the C-terminal half of luciferase in pDuexAC with the sequences encoding the N-terminal half of Venus in pDEST–$^{GW}$VYNE or the C-terminal half of super cyan fluorescent protein in pDEST–$^{GW}$SCYCE[45]. The 4-kb fragments resulted from partial digestion of pDEST–$^{GW}$VYNE and pDEST–$^{GW}$SCYCE by HindIII and EcoRI were ligated into the HindIII/EcoRI site of pDuexAc6. The resulting vectors, pDuvyNE and pDuScyCE, respectively, were used in the LR reaction-based construction of expression vectors. The vectors constructed in this study are listed in Supplementary Table 6.

**Affinity purification–mass spectrometry.** AP–MS was conducted by expressing target proteins fused with a C-terminal GFP tag in the PSB-D Arabidopsis cell culture line using the published protocol[39]. Tandem GFP fused with an N-terminal mitochondrial targeting peptide was used as a negative control. PSB-D cells were cultured in the dark at 25 °C with shaking at 120 r.p.m. using Murashige and Skoog Basal Salts with minimal organics medium (MSMO; Sigma-Aldrich, Gillingham, UK) supplemented with $50 \mu g l^{-1}$ kinetin, $0.5 mg l^{-1}$ 1-naphthaleneacetic acid and 3% sucrose. Cells were subcultured every week at a 1–10 split. Agrobacterium tumefaciens strain GV3101 pMP90 transformed with an Expression vector was grown on a plate for 2 days and then scratched and re-suspended into MSMO medium to gain $OD_{600}$ of 1.0. A 3 ml aliquot of 2-day-old PSB-D cell culture was mixed with 200 μl of A. tumefaciens suspension and 6 μl of 100 mM acetosyringone and co-cultivated for 48 h. Transformed cells were selected in a medium containing $25 \mu g ml^{-1}$ of kanamycin, $500 \mu g ml^{-1}$ carbenicillin and $500 \mu g ml^{-1}$ vancomycin for three rounds of 1-week subculture followed by those with medium containing only kanamycin for two rounds. Expression and localization of the tagged proteins were evaluated by viewing GFP fluorescence using confocal microscopy. The transformed cells were collected by vacuum filtration at 5 days after subculturing and frozen in liquid nitrogen. After grinding into a fine powder using a ball mill (MM301, Retch, Haan, Germany), proteins were extracted by mixing 2 g of material with extraction buffer (25 mM Tris-HCl pH 7.5, 15 mM $MgCl_2$, 5 mM EGTA, 0.1% (v/v) NP40, 1 mM dithiothreitol, 5% (v/v) ethylenglycol and 1 mM phenylmethylsulfonyl fluoride). Following removal of cell debris by repeated centrifugation at 22,000 g, 4 °C for 5 min, the supernatant was mixed with 25 μl of GFP-Trap_A slurry (ChromoTek, Martinsried, Germany) equilibrated with extraction buffer and incubated for 1 h at 4 °C with rotation. The beads were collected by centrifugation at 3,000 g at 4 °C for 3 min and washed three times each with extraction buffer containing 0, 250 and 500 mM of NaCl. The proteins remaining on the beads were subsequently subjected to in-solution digestion by LysC and trypsin and the resulting peptides were purified by a C18 tip[70]. LC-MS/MS analysis was performed on Q Exactive Plus (Thermo Fisher Scientific). Quantitative analysis of MS/MS measurements was performed with the Progenesis QI software (Nonlinear Dynamics, Newcastle, UK). Proteins were identified from spectra using Mascot (Matrix Science, London, UK). Mascot search parameters were set as follows: TAIR10 protein annotation, requirement for tryptic ends, one missed cleavage allowed, fixed modification: carbamidomethylation (cysteine), variable modification: oxidation (methionine), peptide mass tolerance = ± 10 p.p.m., MS/MS tolerance = ± 0.6 Da, allowed peptide charges of + 2 and + 3. A decoy database search was used to limit false discovery rates to 1% on the protein level. Peptide identifications below rank one or with a Mascot ion score below 25 were excluded. Mascot results were imported into Progenesis QI, quantitative peak area information extracted and the results exported for data plotting and statistical analysis. The raw intensities of all detected peptides were shown in Supplementary Data 2. The possible interactions were scored as FC-A score calculated by the SAINT algorithm embedded in the CRAPome website[71,72]. Affinity purification was performed twice and the FC-A values were calculated for the individual replicates.

**Split-LUC complementation assay.** The plasmids for split-LUC assays were extracted from bacterial cells by alkaline lysis and purified by using silicon dioxide (Sigma-Aldrich) slurry[73]. Mesophyll protoplasts were generated from the leaves of Arabidopsis Col-0 by the Tape-Arabidopsis Sandwich method[74]. Briefly, a lower epidermal surface of a leaf was removed by peeling with a strip of tape fixed to it. The mesophyll cells remaining on the tape were incubated in 20 mM 2-(N-morpholino)ethanesulfonic acid (MES) buffer (pH 5.7) containing 1% cellulose (Yakult, Tokyo, Japan), 0.25 % macerozyme (Yakult), 10 mM $CaCl_2$, 20 mM KCl, 0.1% BSA and 0.4 M mannitol with gentle agitation for 20–60 min until the protoplasts were released into the solution. The protoplasts were washed twice with W5 solution (2 mM MES pH 5.7, 154 mM NaCl, 125 mM $CaCl_2$, 5 mM KCl, 5 mM glucose), incubated on ice for 30 min, centrifuged and resuspended into MMg

solution (4 mM MES pH 5.7, 15 mM MgCl₂, 0.4 M mannitol). Protoplasts were transfected with plasmid in a U-bottom 96-well plate by incubating for 5 min at room temperature under the presence of 20% (w/v) PEG4000. Following washing with W5 solution twice, the protoplasts were incubated in the dark at 25 °C overnight[68]. For luminescence detection, 10 µl of 60 µM ViviRen Live Cell substrate (Promega, Madison, WI) was added to each well of a 96-well plate[68]. The plate was kept for 4 min at room temperature in the dark before measuring the luminescence using a CLARIOstar microplate reader (BMG LABTECH, Ortenberg, Germany) with 10 s integration periods at emission 480 ± 30 nm. The experiments were repeated four times.

**Y2H assay.** The expression clones for Y2H were transformed into yeast mating strains AH109 and Y187 by lithium acetate method. Binary protein interactions were tested by direct mating of a set of baits with a set of preys expressed in opposite yeast mating types[69,75]. Diploids containing both bait and prey constructs were inoculated on synthetic dextrose plates without Leu, Trp and His containing 3-aminotriazole[69] and colony formation was scored following 3, 5 and 8 days.

**Bimolecular fluorescence complementation.** BiFC constructs were expressed in mesophyll protoplasts as described above. The protoplasts were incubated with MitoTracker orange CMTMRos (Thermo Fisher Scientific) for mitochondrial staining at 37 °C for 10 min followed by 26 °C for 20 min. Confocal images were taken using a DM6000B/SP5 confocal laser scanning microscope (Leica Microsystems, Wetzlar, Germany). BiFC and MitoTracker fluorescence were imaged with a 488 and 555 nm laser excitation, and emission fluorescence was captured by 500–520 and 560–580 nm band-pass emission filters, respectively.

**Combined protein–protein interaction data sets.** We aimed at combining seven data sets (that is, $2 \times$ AP–MS, $4 \times$ split-LUC and $1 \times$ Y2H) containing the protein–protein interactions of 38 proteins. The three methods result in non-symmetric square (38 × 38) matrices, $X_i$, $1 \leq i \leq 7$, whereby the interaction of protein $A$ to protein $B$ does not imply the interaction of protein $B$ to protein $A$ (Supplementary Data 3). Assuming that protein–protein interaction is symmetric, the first step in combining the seven data sets entailed symmetrization. To symmetrize $X_i$, $1 \leq i \leq 7$, we applied the transformation that creates seven symmetric matrices, $S_{i,jk} = \max(X_{i,jk}, X_{i,kj,})$. We next determined the mean of the replicates (4 for luciferase, 2 for AP–MS), denoted by $\overline{S_i}$, normalized to the range [0,1]. This resulted in three final matrices for the three experimental approaches. With these matrices, we apply an approach similar to STATIS[76], whereby we first determined the $R_V$ coefficient for each pair of matrices $\overline{S_i}$ and $\overline{S_j}$, $1 \leq i, j \leq 3$, gathered in the cosine matrix $C$. We use the entries $\alpha_i$ of the first eigenvector of $C$, after re-scaling to unit-norm, as weights to combine the matrices $\overline{S_i}$. This results in the compromise matrix

$$S = \sum_{i=1}^{3} \alpha_i \cdot \overline{S_i},$$

which we report as the final matrix representative of all the considered mean-symmetrized data sets (the larger the value of $\alpha_i$, the bigger the contribution of the respective data set to the compromise matrix). The resulting matrix $S$ was also normalized by dividing with its maximum value. The calculations were carried out in R. For the network calculations we used the igraph R package. The protein–protein interaction network was visualized by Cytoscape[77] using the data shown in Supplementary Table 3.

**Isotope dilution experiments.** *Solanum tuberosum* (cv. Desirée) tubers were purchased from a local supermarket. Mitochondria were isolated on a stepped gradient of Percoll (GE Healthcare Life Sciences, Chicago, IL) consisting of steps of 50, 28 and 20% (v/v) from the 28%/50% Percoll interface. Mitochondria were further purified on a second self-forming Percoll gradient consisting of 28% Percoll[78]. Isolated mitochondria were incubated in a buffer containing 0.1 M MOPS (pH 7.2), 5 mM MgCl₂, 0.2 M mannitol, 0.1% (w/v) BSA, 20 mM KH₂PO₄, 0.3 mM NAD⁺, 0.2 mM ADP, 0.1 mM thiamine pyrophosphate, 0.15 U ml⁻¹ hexokinase, 20 mM glucose and either 10 mM [3-¹³C]pyruvate (Cambridge Isotopes, Anderstown, MA), 10 mM malate and 10 mM malonate or 10 mM [3-¹³C]glutamate (Sigma-Aldrich) 10 mM pyruvate and 0.5 µM fluorocitrate for testing citrate/2OG or succinate/fumarate/malate channelling, respectively. Following 80 min of incubation, when the ¹³C-label accumulation in the product had reached a steady state (Supplementary Fig. 5), unlabelled intermediate (citrate, 2OG, succinate, fumarate or malate) was added to a final concentration of 0.1 mM. The reaction was quenched at 10, 15, 20, 25, 30 and 35 min following the addition of intermediate by incubating a 1 ml aliquot of mitochondrial suspension with 10 mM 1,2,3-benzenetricarboxylate on ice for 5 min. The mitochondria were collected by vacuum filtration on a 0.22 µm-pore membrane filter (GVWP02500; Millipore, Watford, UK), washed with 5 ml of buffer containing 10 mM 1,2,3-benzenetricarboxylate and snap-frozen. Polar metabolites were extracted in 1.46 ml of methanol supplemented with 12 µg of ribitol as an internal standard by incubating for 10 min at 70 °C with shake at 950 r.p.m. Following a centrifugation at 1,100 g, the supernatant was mixed with 750 µl of chloroform and 1.5 ml of H₂O, and centrifuged at 2,200 g. An aliquot of 150 µl of the polar aqueous phase was

dried in a vacuum concentrator and derivatized with 40 µl of 20 mg ml⁻¹ methoxyamine hydrochloride in pyridine for 2 h at 37 °C and 70 µl of N-methyl-N-(trimethylsilyl)trifluroacetamide for 30 min at 37 °C. Label accumulation in metabolites was evaluated by gas chromatography–mass spectrometry according to Williams et al.[79] Briefly, gas chromatography–mass spectrometry analysis was carried out using a 7890A GC (Agilent, Santa Clara, CA) coupled to a 5975C quadrupole mass spectrometer (Agilent). The sample (1 µl) was injected in splitless mode and separated on a HP5-ms column (30 m, 0.25 mm i.d., Agilent). Peak identification was performed using AMDIS32 (National Institute of Standards and Technology, Gaithersburg, MD) and the analysis of ¹³C incorporation was carried out using Chemstation (MSD version; Agilent). Averaged mass spectra were corrected for the presence of naturally occurring heavy isotopes using the MSCorr software[80]. The ratio of isotopologs in the fragments of the product metabolites at each time point is shown in Supplementary Table 7. Fractional carbon enrichment was calculated by dividing the sum of numbers of ¹³C atom (that is, the percentage of isotopolog multiplied by the number of ¹³C in it) with the sum of all carbon atoms in the fragment.

**Data availability.** The authors declare that all data and R scripts supporting the findings of this study are available within the manuscript and its supplementary files or are available from the corresponding author upon request.

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

## Acknowledgements

This work was supported by funding from EU FP7 Project ERA-NET Plant Genomics, STRESSNET (L.J.S., A.R.F., T.O.), German Research Foundation (DFG; T.O.), the Max-Planck Society (Z.N., A.G., A.R.F., T.O.), German Academic Exchange Service (DAAD; Y.Z.), International Max Planck Research Schools PhD programme (IMPRS; Y.Z.) and BBSRC CASE studentship with Advanced Technologies, Cambridge, UK (K.F.M.B.). We thank Dr Geert de Jaeger and Dr Jelle Van Leene (VIB, Belgium) for providing the PSB-D cell culture, Dr Naohiro Kato (Louisiana State University, LA) for providing the split-

LUC vectors, Dr Peter Uetz (Karlsruhe Institute of Technology, Germany) for providing yeast-two-hybrid vectors, Juliane Neupert (MPIMP) and Dr Hardy Chan (Charité-Universitätsmedizin, Berlin, Germany) for their help in luminescence detector and Eugenia Maximova (MPIMP) for her help with the confocal microscopy.

## Author contributions

Y.Z., R.G.R., L.J.S., A.R.F. and T.O. designed experiments. Y.Z., S.B. and T.O. performed cDNA cloning and vector construction. Y.Z., I.K. and T.O. developed and conducted protein–protein interaction assays. C.S. and A.G. performed the LC-MS/MS analysis. K.F.M.B., R.G.R. and L.J.S. performed the isotope dilution experiments. Z.N. and T.O. analysed the data. R.G.R. and L.J.S. performed critical reading and contributed to manuscript writing. A.R.F. and T.O. wrote the manuscript.

## Additional information

**Competing interests:** The authors declare no competing financial interests.

**Publisher's note**: 

