## [Peer Review File · Nature Communications]

Reviewers' comments:

Reviewer #1 (Remarks to the Author):

The authors utilize interactomics and substrate channeling assays to investigate the TCA metabolon in plants. Although the TCA metabolon has been studied frequently in other organisms, it has not been studied in plants, so this work is original and of interest to plant scientists as well as metabolic scientists. The data and methodology is valid and the authors appropriately used statistics and appropriate control experiments. The references are representative of the field and the paper is well written and concise.

I find this is a very interesting paper that will be of broad interest to the field, but there are several issues that need to be addressed:

1. The authors do structural studies with *Arabidopsis* and channeling studies with *Solanum*. They should utilize the same organism for both studies in order to be making appropriate comparisons.
2. Although the authors utilized good controls and use of statistics, the paper could benefit from more quantitative discussion.
3. All mass spectrometry-based interactomic techniques have different advantages and disadvantages. The authors should more clearly explain their choice of technique and its advantages and disadvantages.
4. Generally, it is best to verify substrate channeling with more than one method. I would suggest adding an additional method to the analysis for thoroughness.
5. Generally, the authors make linearizing the cycle appear straight forward, but it is more complex than this in-vivo and should be explained in more detail.

Reviewer #2 (Remarks to the Author):

- The key result of this paper is the demonstration of the existence of channelling in TCA cycle in plants. They first demonstrate the existence of protein-protein interactions between proteins of TCA cycle in plants. The functional consequences of these protein complexes at the level of metabolic channelling of TCA cycle intermediates are also investigated using ¹³C-labelled substrates and GC-MS technique.

- Physical interactions between tricarboxylic acid cycle enzymes has been already described for other organisms as in *Bacillus subtilis* and indirect evidences for a metabolon has been provided. Protein-protein interactions map in plant TCA cycle has not been explored systematically using different techniques, so in this aspect the results obtained are novel.

- Methodologies used for protein-protein interactions are appropriate and enough cross-validation with different techniques is provided to sustain the conclusions on protein-protein interactions identified.

¹³C-based methods are considered appropriate to investigate metabolic channelling. The obtained results are indicative of the existence of metabolic channelling in citrate, fumarate and malate in TCA cycle in plants. However, more control data needs to be provided to unequivocally conclude that metabolic channelling exists.

Next major concerns, suggested improvements, and additional data required to

demonstrate metabolic channelling are listed:

1) To reach isotopic steady-state in TCA cycle intermediates has been reported to require longer time periods in intact cells. In the manuscript it is reported that, in isolated mitochondria, isotopic steady-state is achieved in 80 minutes after incubation of mitochondria in the presence of ^{13}C -labelled substrates. The following incubations adding cold metabolites are performed for 35 minutes. A time-course demonstrating that isotopic steady-state of TCA cycle intermediates is achieved at 80 minutes after incubation with the labelled substrate and that this steady-state is maintained during at least 45 minutes more, is necessary as a control. This time-course should be included in the supplementary material.

2) The authors use inhibitors of key enzymes of TCA cycle to ensure that the pathway is lineal. Control experiments need to be done to demonstrate that mitochondria remain fully functional and metabolically active in the presence of these inhibitors. For example:

a. Data on quantitative uptake of labelled and cold-metabolites under the different experimental conditions are not reported. These data are necessary to quantify the metabolic activity of the mitochondria and the quantitative importance of the channelling. Measurement of mitochondrial activity (seahorse measurements or similar) need to be done to be sure that mitochondrial metabolism is active in presence of the inhibitors during all the time courses with added cold metabolites

b. It is necessary to demonstrate that, in the presence of malonate, pyruvate continues being metabolized. For this purpose a control adding cold pyruvate and following the label in succinate would be needed

3) The % of decrease in label enrichment that has been considered for justifying the non-existence of channelling is around 0.02 or 0.03 in the best-case scenario. A 0.02 or 0.03% decrease in label enrichment is very low taking into account that the method requires derivatization of the metabolites and data correction using MSCorr software. A difference of 0.02% in the enrichment can perfectly be due to experimental error or variation, and is probably not statistically significant. For instance, in Figure 4e it is observed a decrease on citrate enrichment of around 0.02% due to the addition of cold succinate (after 30 or 35 minutes of incubation) which is considered indicative that succinate is not channelled. However, the label in citrate suffered an increase of similar magnitude after incubation with cold fumarate (Figure 4f) that is sustained during all the incubation period. The authors do not give a convincing explanation for the increase of citrate label after incubation with cold fumarate. Moreover, a little increase in citrate enrichment is also observed when mitochondria were incubated with cold malate, increase that is also not justified (Figure 4g). Thus, a similar absolute change (of 0.02%) in the label of citrate is interpreted as non-channelling (in the case of the enrichment decrease observed after succinate addition) or as negligible (in the case of the enrichment increase observed after fumarate addition). If channelling exists it would be expected that label enrichment is maintained constant during all the incubation with cold intermediary metabolites. A convincing explanation for label enrichments needs to be provided.

4) The % of the m_1 isotopologue of citrate and other measured metabolites needs to be provided at the different time points. It is important to know which is the initial m_1 value in the succinate and citrate pools (prior to the addition of the cold metabolites) and also during the time courses. The differential enrichment is not enough to evaluate the robustness of these experimental data.

5) Regardless of linearization of TCA cycle is achieved using inhibitors, it is not demonstrated that other branches in the cycle are not active. For instance, malate can be converted into pyruvate through malic enzyme, and this pyruvate can be converted to alanine through mitochondrial alanine transaminase. If this happens, malate will not dilute citrate label due to the fact that malate would go to alanine which is not related with the proposed channelled pathway. The same should be applied to fumarate, once it is converted to malate. The scape through the branch can be influenced on the local concentration, transport rates etc... Moreover, 2OG can be converted into glutamate through the reversible enzyme glutamate dehydrogenase, thus affecting to the supposed channelling in citrate.

Reviewer #3 (Remarks to the Author):

A. Summary of the key results

Zhang et al. studied the formation of supramolecular complexes of sequential metabolic enzymes (aka metabolon) in plant tricarboxylic acid (TCA) cycle. Three independent approaches were carried out to test and score protein-protein interactions: affinity purification-mass spectrometry, split-luciferase assay, and yeast-two-hybrid assay. A total of 158 interactions between subunits of the enzymes of the TCA cycle were identified, including novel interactions between non-adjacent reactions. Selected protein-protein interactions were confirmed by bimolecular fluorescent complementation in Arabidopsis mesophyll protoplasts. Finally, ¹³C-isotopic dilution experiments were conducted in mitochondria isolated from potato to tentatively provide evidence for metabolite channeling. It was concluded that citrate, malate and fumarate were indeed channelled whereas 2-oxoglutarate and succinate were not channelled, which was in agreement with the interactome study.

B. Originality and interest: if not novel, please give references

The novelty of this work does not rely on the methodologies used to study the protein-protein interaction but rather in the combination of three different approaches to decrease the number of false-positives, and capture weak interactions. Several interactions between subunits within an enzymatic complex or between subunits from sequential reactions were demonstrated to occur in TCA cycle in other organisms. However, this study provides novel information on the plant TCA cycle, as new interactions between enzymes from adjacent and non-adjacent reactions were identified.

C. Data & methodology: validity of approach, quality of data, quality of presentation

The protein-protein interactions were tested using three approaches: affinity purification-mass spectrometry (n=2), split-luciferase assay (n=4), and yeast-two-hybrid assay (n=1). Overall the methods and the data processing are well described. However, it is not clear if the replicates were treated independently to determine the scores (page 4 last paragraph) or if they were averaged beforehand. For instance, the values for the dataset #1 for AP-MS are not in the same range than the dataset #2; how was that accounted for? Same question for the four replicates of the split-luciferase assays.

Regarding the isotopic dilution experiments, there are several points of concern: 1) the GC-

MS method that is referred is a metabolomics profiling approach and not a method to determine ^{13}C -labeling. If not described and validated elsewhere, authors must provide a table with the mass isotopomers that were followed for each derivatized organic acid, the number of derivative groups, the sensitivity/precision as well as a validation of the method (using organic acids at a known % labeling). 2) Fig 4 only reports differences in fractional enrichment of label; labeling data must be provided as a supplemental table. 3) differences in labeling were found to be in the order of 0.02%, which is extremely low considering the natural abundance of ^{13}C (1.1%). What is the precision for the determination of the labeling? 4) the labeling was measured in duplicated samples, but three biological replicates are commonly required in this field. 5) more data on the labeling of the other organic acids must be included besides the labeling for succinate and citrate in order to provide evidence for metabolite channeling. For instance, when adding unlabeled citrate, the result obtained in Fig 4b could be explained by a slow/non uptake of citrate (and so on for the other experiments). 6) Fig 4 e,f,g show an increase of labeling in citrate after addition of unlabeled organic acids, please explain/discuss this rather unexpected result.

D. Appropriate use of statistics and treatment of uncertainties

Not applicable

E. Conclusions: robustness, validity, reliability

The biophysical evidence for protein-protein interaction is robust: most studies are using one or two approaches whereas this one successfully combines and scores three methods. Several interactions have been validated by bimolecular fluorescent complementation, showing that the "compromise" approach is reliable.

There are several concerns about the experiments testing the metabolic channeling (see comments in section C). It is therefore difficult to judge the reliability of the conclusions from this section.

F. Suggested improvements: experiments, data for possible revision

Please see comments in section C.

One of the novel findings of this manuscript is the interaction between subunits from the malic enzyme and pyruvate dehydrogenase. Labeling-dilution testing the metabolic channeling of malate into Acetyl-CoA would be appreciated.

The authors state that no labeling-dilution experiment was attempted with isocitrate because it is not uptaken by isolated mitochondria. Rasmussen and Moller (Plant Physiol. 1990) measured isocitrate oxidation and corresponding respiration in isolated potato mitochondria, which proves that isocitrate is taken up by mitochondria. Isotopic dilution experiment with unlabeled isocitrate would be very valuable to test metabolic channeling. The interactome study was based on proteins from Arabidopsis whereas the isotope dilution experiments were conducted in isolated potato mitochondria. Please discuss: i) how similar/different the protein sequences are between the two species; ii) the limitations of this study (can we assume that isolated potato mitochondria behave like other plant mitochondria?); iii) the flux around the TCA cycle was found to be absent in developing Brassica embryos (Schwender et al., JBC 2006), how does this finding compare to the present study?

G. References: appropriate credit to previous work?

78 references are currently listed, and they adequately credit previous work.

H. Clarity and context: lucidity of abstract/summary, appropriateness of abstract, introduction and conclusions

Overall, the manuscript is well written. The abstract has a concise description of the study and the main findings. The introduction clearly presents the state of the knowledge about metabolons, evidence of metabolons, metabolons in TCA cycle demonstrated in other organisms...

Reviewer #4 (Remarks to the Author):

This manuscript by Zhang et al is an interesting and unique investigation carried out by employing multiple techniques to find out the interactome of TCA cycle in Arabidopsis.

While going through the manuscript I felt the need to see the mass Spectrometry data of the study. I encourage authors to submit the MS data as suppl. file. Without this data it is impossible to check the quality of the Affinity purification-Mass Spectrometry (AP-MS) especially when the authors have used MS after ONLY one step of purification. My experience is that two-step purifications are better for MS.

Moreover, all the techniques of this manuscript such as cell culture, Y2H, split-LUC - all of them can be considered as in-vitro, at least technically speaking none is in-planta approach. Authors should give a justification why they preferred cell culture over transgenic Arabidopsis plants for the AP-MS.

Suppl. table legends need more information on what the number in different columns mean?

Check the manuscript for typos. One example is Table S6 - "Doner"

Reviewers' comments:

Reviewer #1 (Remarks to the Author):

The authors utilize interactomics and substrate channeling assays to investigate the TCA metabolon in plants. Although the TCA metabolon has been studied frequently in other organisms, it has not been studied in plants, so this work is original and of interest to plant scientists as well as metabolic scientists. The data and methodology is valid and the authors appropriately used statistics and appropriate control experiments. The references are representative of the field and the paper is well written and concise.

I find this is a very interesting paper that will be of broad interest to the field, but there are several issues that need to be addressed:

1. The authors do structural studies with *Arabidopsis* and channeling studies with *Solanum*. They should utilize the same organism for both studies in order to be making appropriate comparisons.

We agree that it would be ideal to use the same organism for both studies but it was impossible for the following reasons. The interactome approach requires very well annotated genomic information to identify all the genes involved in mitochondrial TCA cycle reliably. In addition the AP-MS protocol has been very well established in *Arabidopsis* and this enabled us to combine three independent (semi)quantitative techniques. For these reasons the structural studies could only be conducted with *Arabidopsis*. On the other hand, the quality and quantity of mitochondria isolated from *Arabidopsis* are not high enough to conduct channeling studies. In spite of this limitation, our results showed highly conserved metabolon composition even among kingdoms of life and the difference between plant species is expected to be minor. In addition, the comparison of amino acid sequences of *Arabidopsis* and potato protein showed high similarity between the TCA cycle enzyme subunits from two organisms, suggesting similar enzymatic properties in these two species. These points have been mentioned on P11 L5-7 and P6 L24-27.

2. Although the authors utilized good controls and use of statistics, the paper could benefit from more quantitative discussion.

Quantitative analysis of our results potentially leads to the information related to the stability of interaction and its relationship to the metabolite channeling, which is one of the most important aspects to establish the functions of dynamic metabolon formation in metabolic regulation. However, we did not include quantitative discussion relay on the compromise score in order to avoid possibly misleading conclusions since the relationship between a compromise score and stability of the protein-protein interaction has not been proven. The combination of three methodologies is expected to capture the interactions which are preferentially detected by particular methods and increase the chance of detecting weak interactions. However we cannot exclude the possible effects of preferential detection on the score, which can compromise its linear relationship against stability of the interaction. It might be possible to prove this relationship by correlating our compromise scores to the results of biophysical analyses such as surface plasmon resonance and microscale thermophoresis. However, this comparison is out of the main scope of this study. We consider our compromise score is rather related to the reliability of the detection and used it only for selecting the most provable interactions. To clarify this, the word "stability" (of detection) has been removed from P5L7.

We decided to include no quantitative discussion on the isotope dilution experiments since there are too many unknown factors which probably affect the degree of isotope dilution including the rate of intermediate uptake, concentration of the intermediate in the mitochondria, catalytic efficiencies of individual isozymes and their abundance.

3. All mass spectrometry-based interactomic techniques have different advantages and disadvantages. The authors should more clearly explain their choice of technique and its advantages and disadvantages.

We employed single step purification for AP-MS analysis to detect weak interactions which might be lost with multiple washing processes in tandem-affinity purification. The disadvantage of low specificity can be compensated by combining with other techniques. Two sentences describing the reason for the choice of technique have been added on P8 L3-8.

4. Generally, it is best to verify substrate channeling with more than one method. I would suggest adding an additional method to the analysis for thoroughness.

We agree that the use of multiple methods is the best way to establish substrate channeling. However, for the *in vivo* (in organelle) measurement there are only two available methods: isotope dilution and orientation conserved label transfer. The latter relies on the symmetric structure of fumarate and succinate restricting its application to the channeling of these molecules. Indeed we have tried this method with isolated potato mitochondria but it is technically complicated due to the effects of divalent cations on the NMR analysis. In contrast the isotope dilution method used here is a well-established method with a much wider range of application. A description regarding the limitation of orientation conserved label transfer has been added on P8 L36-41.

5. Generally, the authors make linearizing the cycle appear straight forward, but it is more complex than this in-vivo and should be explained in more detail.

The simultaneous application of an inhibitor and a ^{13}C -labeled TCA cycle intermediate as substrate leads to continuous accumulation of the labeled product, which is the substrate of the inhibited enzyme. A more detailed description of the use of inhibitors to linearize the TCA cycle in isolated mitochondria has been added on P6 L37-P7 L9. The possible effects of branching pathways have also been discussed on P7 L25-32.

Reviewer #2 (Remarks to the Author):

- The key result of this paper is the demonstration of the existence of channelling in TCA cycle in plants. They first demonstrate the existence of protein-protein interactions between proteins of TCA cycle in plants. The functional consequences of these protein complexes at the level of metabolic channelling of TCA cycle intermediates are also investigated using ^{13}C -labelled substrates and GC-MS technique.

- Physical interactions between tricarboxylic acid cycle enzymes has been already described for other organisms as in *Bacillus subtilis* and indirect evidences for a metabolon has been provided. Protein-protein interactions map in plant TCA cycle has not been explored systematically using different techniques, so in this aspect the results obtained are novel.

- Methodologies used for protein-protein interactions are appropriate and enough cross-

validation with different techniques is provided to sustain the conclusions on protein-protein interactions identified.

¹³C-based methods are considered appropriate to investigate metabolic channelling. The obtained results are indicative of the existence of metabolic channelling in citrate, fumarate and malate in TCA cycle in plants. However, more control data needs to be provided to unequivocally conclude that metabolic channelling exists.

These have been added as requested (detailed descriptions of the changes are provided below).

Next major concerns, suggested improvements, and additional data required to demonstrate metabolic channelling are listed:

1) To reach isotopic steady-state in TCA cycle intermediates has been reported to require longer time periods in intact cells. In the manuscript it is reported that, in isolated mitochondria, isotopic steady-state is achieved in 80 minutes after incubation of mitochondria in the presence of ¹³C-labelled substrates. The following incubations adding cold metabolites are performed for 35 minutes. A time-course demonstrating that isotopic steady-state of TCA cycle intermediates is achieved at 80 minutes after incubation with the labelled substrate and that this steady-state is maintained during at least 45 minutes more, is necessary as a control. This time-course should be included in the supplementary material.

A new supplementary figure (Supplementary Figure 5) has been added that shows these controls. The new data demonstrate that isotopic steady state is maintained during the relevant time period. This result has been stated on P7 L2-6.

2) The authors use inhibitors of key enzymes of TCA cycle to ensure that the pathway is lineal. Control experiments need to be done to demonstrate that mitochondria remain fully functional and metabolically active in the presence of these inhibitors. For example:

a. Data on quantitative uptake of labelled and cold-metabolites under the different experimental conditions are not reported. These data are necessary to quantify the metabolic activity of the mitochondria and the quantitative importance of the channelling. Measurement of mitochondrial activity (seahorse measurements or similar) need to be done to be sure that mitochondrial metabolism is active in presence of the inhibitors during all the time courses with added cold metabolites b. It is necessary to demonstrate that, in the presence of malonate, pyruvate continues being metabolized For this purpose a control adding cold pyruvate and following the label in succinate would be needed

We provide the following new data to demonstrate that the mitochondria remain fully functional and active in the presence of the inhibitors. First, we show that succinate accumulates steadily during a time course when isolated mitochondria were incubated with labelled pyruvate, unlabeled malate and malonate, demonstrating that pyruvate continues to be metabolized in the presence of the inhibitor (Supplementary Figure 3a). Similarly, citrate accumulates when mitochondria are provided with labelled glutamate, unlabeled pyruvate and fluorocitrate, demonstrating that glutamate is metabolized in the presence of the inhibitor (Supplementary Figure 3b). In addition, we used ³¹P NMR to measure the rate of synthesis of ATP in the presence of inhibitors. As is expected, linearization of the pathway reduced the rate of ATP synthesis because the linearized pathways generate less reducing power per mole of substrate catabolized. However, ATP was still synthesized at an appreciable rate (around 50 nmol min⁻¹ mg protein⁻¹ during a 1 h period in the presence of malonate and fluorocitrate (Supplementary

Figure 4). This demonstrates that both carboxylic acid metabolism and electron transport are active and functional in the presence of the inhibitors. The sentences describing these results have been added on P6 L39-P7 L2.

3) The % of decrease in label enrichment that has been considered for justifying the non-existence of channelling is around 0.02 or 0.03 in the best-case scenario. A 0.02 or 0.03% decrease in label enrichment is very low taking into account that the method requires derivatization of the metabolites and data correction using MSCorr software. A difference of 0.02% in the enrichment can perfectly be due to experimental error or variation, and is probably not statistically significant. For instance, in Figure 4e it is observed a decrease on citrate enrichment of around 0.02% due to the addition of cold succinate (after 30 or 35 minutes of incubation) which is considered indicative that succinate is not channelled. However, the label in citrate suffered an increase of similar magnitude after incubation with cold fumarate (Figure 4f) that is sustained during all the incubation period. The authors do not give a convincing explanation for the increase of citrate label after incubation with cold fumarate. Moreover, a little increase in citrate enrichment is also observed when mitochondria were incubated with cold malate, increase that is also not justified (Figure 4g). Thus, a similar absolute change (of 0.02%) in the label of citrate is interpreted as non-channelling (in the case of the enrichment decrease observed after succinate addition) or as negligible (in the case of the enrichment increase observed after fumarate addition). If channelling exists it would be expected that label enrichment is maintained constant during all the incubation with cold intermediary metabolites. A convincing explanation for label enrichments needs to be provided.

We apologize for the confusing way in which the data in Fig 4 were presented. We were using a fractional enrichment scale where 1.0 equals 100% labelling and presented the percentage difference between the fractional enrichment before and after the addition of the indicated unlabeled intermediates. We also noticed an error in the way in which fractional enrichment had been calculated. We have recalculated the data on a more conventional % scale for fractional enrichment and the graphs now show the absolute difference in fractional enrichment, not percentage difference. We also noticed that we had taken the wrong data point as the 'before addition of unlabeled intermediate' and have corrected this. The decreases in fractional enrichment due to isotope dilution expressed in this manner are up to 1.5%. These recalculations do not change the conclusions that can be drawn from the experiment. To demonstrate that our measurements are sufficiently precise to reliably quantify such changes, we calculated the precision of the measurement from a time-series where citrate or succinate labelling had reached steady state (Supplementary Figure 5). From these we calculate that succinate can be measured with a precision of 0.14% fractional enrichment and citrate with a precision of 0.06%. The results of these analyses have been added on P7 L2-6 and P7 L21-23. Hence we can reliably quantify the changes in enrichment reported. We also emphasize that the data has been compared to the 95% confidence interval (shown as the grey shading in Figure 4) and where this falls below the 0 line, there is a statistically significant difference in fractional enrichment. This has been made clear in the legend to Figure 4.

4) The % of the m1 isotopologue of citrate and other measured metabolites needs to be provided at the different time points. It is important to know which is the initial m1 value in the succinate

and citrate pools (prior to the addition of the cold metabolites) and also during the time courses. The differential enrichment is not enough to evaluate the robustness of these experimental data. A table showing isotopolog composition in the product metabolites over the experimental period has been added as Supplementary Table S8. Abundance of M+1 isotopolog of product molecules reached 85-90% prior to the addition of the cold metabolites. Negligible alteration in the abundance of isotopologs with more than one ^{13}C was observed over the experimental period, suggesting proper inhibition of second round TCA cycle reactions.

5) Regardless of linearization of TCA cycle is achieved using inhibitors, it is not demonstrated that other branches in the cycle are not active.

For instance, malate can be converted into pyruvate through malic enzyme, and this pyruvate can be converted to alanine through mitochondrial alanine transaminase. If this happens, malate will not dilute citrate label due to the fact that malate would go to alanine which is not related with the proposed channelled pathway. The same should be applied to fumarate, once it is converted to malate. The scape through the branch can be influenced on the local concentration, transport rates etc... Moreover, 2OG can be converted into glutamate through the reversible enzyme glutamate dehydrogenase, thus affecting to the supposed channelling in citrate.

This is an interesting point. With respect to the conversion of 2OG to glutamate by glutamate dehydrogenase, this reaction will not occur here because we have not supplied ammonium. However, the malate branch pathway via malic enzyme could be active in our system. The effect of malic enzyme branch on the isotope dilution resulting from the addition of unlabeled malate is expected to be minor since this pathway produces unlabeled pyruvate which was added to the system in excess. When malate is channeled between fumarase and malate dehydrogenase, ME cannot have access to ^{13}C -malate and no dilution of label in citrate would be observed. The ME branch could prevent the dilution of label accumulation in citrate if a large amount of malate is removed by the ME reaction as the reviewer suggested. This is unlikely since there is a small dilution of label in citrate when unlabeled malate is added (Fig. 4g). Thus the ME branch does not affect our conclusion from the data presented. The other scenario raised by the reviewer is represented by the data obtained after addition of unlabeled fumarate (Fig 4f) where there is no detectable dilution of label in the citrate pool. We concluded this is evidence for metabolite channeling. It is still possible that the unlabeled malate generated by the action of fumarase is metabolized exclusively to alanine and cannot therefore dilute the label in the citrate pool. However, this is unlikely because we know the effect of unlabeled malate is to lead to a small dilution of the citrate pool (as discussed above). A brief discussion on these points has been added to the manuscript (P7 L25-32).

Reviewer #3 (Remarks to the Author):

A. Summary of the key results

Zhang et al. studied the formation of supramolecular complexes of sequential metabolic enzymes (aka metabolon) in plant tricarboxylic acid (TCA) cycle. Three independent approaches were carried out to test and score protein-protein interactions: affinity purification-mass spectrometry,

split-luciferase assay, and yeast-two-hybrid assay. A total of 158 interactions between subunits of the enzymes of the TCA cycle were identified, including novel interactions between non-adjacent reactions. Selected protein-protein interactions were confirmed by bimolecular fluorescent complementation in *Arabidopsis mesophyll* protoplasts. Finally, ¹³C-isotopic dilution experiments were conducted in mitochondria isolated from potato to tentatively provide evidence for metabolite channeling. It was concluded that citrate, malate and fumarate were indeed channeled whereas 2-oxoglutarate and succinate were not channeled, which was in agreement with the interactome study.

B. Originality and interest: if not novel, please give references

The novelty of this work does not rely on the methodologies used to study the protein-protein interaction but rather in the combination of three different approaches to decrease the number of false-positives, and capture weak interactions. Several interactions between subunits within an enzymatic complex or between subunits from sequential reactions were demonstrated to occur in TCA cycle in other organisms. However, this study provides novel information on the plant TCA cycle, as new interactions between enzymes from adjacent and non-adjacent reactions were identified.

C. Data & methodology: validity of approach, quality of data, quality of presentation

The protein-protein interactions were tested using three approaches: affinity purification-mass spectrometry (n=2), split-luciferase assay (n=4), and yeast-two-hybrid assay (n=1). Overall the methods and the data processing are well described. However, it is not clear if the replicates were treated independently to determine the scores (page 4 last paragraph) or if they were averaged beforehand. For instance, the values for the dataset #1 for AP-MS are not in the same range than the dataset #2; how was that accounted for? Same question for the four replicates of the split-luciferase assays.

Means of the data from replicated experiments were used to calculate the compromise score as described in Materials and Methods (P14 L11-12). We agree the AP-MS data showed certain variation in values but the set of proteins detected with higher value was quite similar.

Regarding the isotopic dilution experiments, there are several points of concern: 1) the GC-MS method that is referred is a metabolomics profiling approach and not a method to determine ¹³C-labeling. If not described and validated elsewhere, authors must provide a table with the mass isotopomers that were followed for each derivatized organic acid, the number of derivative groups, the sensitivity/precision as well as a validation of the method (using organic acids at a known % labeling).

We apologize for the inappropriate citation. We have previously established and validated the use of GC-MS for quantification of ¹³C-labelling of organic acids and have now cited the appropriate reference (Williams, T. C. R. *et al.* (2010) A genome-scale metabolic model accurately predicts fluxes in central carbon metabolism under stress conditions. *Plant Physiol* **154**, 311–323). We have also additionally provided new data that quantifies the precision of the measurements (Supplementary Figure 5).

2) Fig 4 only reports differences in fractional enrichment of label; labeling data must be provided as a supplemental table.

Labeling data during pre-incubation and following the addition of non-labeled intermediate in all experiments has been presented as Supplementary Table 8.

3) differences in labeling were found to be in the order of 0.02%, which is extremely low considering the natural abundance of ^{13}C (1.1%). What is the precision for the determination of the labeling?

We apologize for the confusing way in which the data in Fig 4 were presented. We were using a fractional enrichment scale where 1.0 equals 100% labelling and presented the percentage difference between the fractional enrichment before and after the addition of the indicated unlabeled intermediates. We also noticed an error in the way in which fractional enrichment had been calculated. We have recalculated the data on a more conventional % scale for fractional enrichment and the graphs now show the absolute difference in fractional enrichment, not percentage difference. We also noticed that we had taken the wrong data point as the ‘before addition of unlabeled intermediate’ and have corrected this. The decreases in fractional enrichment due to isotope dilution expressed in this manner are up to 1.5%. These recalculations do not change the conclusions that can be drawn from the experiment. To demonstrate that our measurements are sufficiently precise to reliably quantify such changes, we calculated the precision of the measurement from a time-series where citrate or succinate labelling had reached steady state (Supplementary Figure 5). From these we calculate that succinate can be measured with a precision of 0.14% fractional enrichment and citrate with a precision of 0.06%. The results of these analyses have been added on P7 L2-6 and P7 L21-23. Hence we can reliably quantify the changes in enrichment reported. We also emphasize that the data has been compared to the 95% confidence interval (shown as the grey shading in Figure 4) and where this falls below the 0 line, there is a statistically significant difference in fractional enrichment. This has been made clear in the legend to Figure 4.

4) the labeling was measured in duplicated samples, but three biological replicates are commonly required in this field.

Due to the large amount of isolated mitochondria required for the isotope dilution experiments, it was not feasible to make three biological replicates for all experiments. Indeed each experiment required multiple separate mitochondrial preparations (each taking half a day to prepare) to be pooled together. We note that we were not seeking to make quantitative assessments of the degree of channeling, but were rather looking for a more qualitative assessment of the presence of channeling. Moreover, the fact that each experiment consisted of pools of separate mitochondria means that there is a high degree of replication within each experiment which captures the biological variation. Hence a qualitatively consistent result in two independent experiments can be considered highly significant.

5) more data on the labeling of the other organic acids must be included besides the labeling for succinate and citrate in order to provide evidence for metabolite channeling. For instance, when adding unlabeled citrate, the result obtained in Fig 4b could be explained by a slow/non uptake of citrate (and so on for the other experiments).

This explanation is not likely since it is well established that potato mitochondria take up and metabolize all of the tested organic acids. We are also not clear how analysis of labelling of organic acids other than citrate and succinate would address this possibility – since if there is metabolite channeling then we would not see dilution of these labelled pools. In any case, the

concentrations of other TCA cycle organic acids were too low to reliably quantify labelling – this was only possible for citrate and succinate which accumulated to high levels due to the linearization of the cycle to make these two metabolite pathway endpoints.

6) Fig 4 e,f,g show an increase of labeling in citrate after addition of unlabeled organic acids, please explain/discuss this rather unexpected result.

This was due to an error in data processing (the wrong time point was taken as 'zero'). This has now been corrected.

D. Appropriate use of statistics and treatment of uncertainties

Not applicable

E. Conclusions: robustness, validity, reliability

The biophysical evidence for protein-protein interaction is robust: most studies are using one or two approaches whereas this one successfully combines and scores three methods. Several interactions have been validated by bimolecular fluorescent complementation, showing that the "compromise" approach is reliable.

There are several concerns about the experiments testing the metabolic channeling (see comments in section C). It is therefore difficult to judge the reliability of the conclusions from this section.

These concerns have been allayed (see the responses to the comments in section C above).

F. Suggested improvements: experiments, data for possible revision

Please see comments in section C.

One of the novel findings of this manuscript is the interaction between subunits from the malic enzyme and pyruvate dehydrogenase. Labeling-dilution testing the metabolic channeling of malate into Acetyl-CoA would be appreciated.

It would indeed be interesting to test pyruvate channeling between malic enzyme and pyruvate dehydrogenase. However, this is technically difficult since acetyl-CoA is extremely difficult to quantify by GC-MS and other MS techniques. Following the labelling into a downstream product such as citrate does not lead to a clear resolution of channeling due to the complication of the branch point at malate (see Fig 4d).

The authors state that no labeling-dilution experiment was attempted with isocitrate because it is not uptaken by isolated mitochondria. Rasmussen and Moller (Plant Physiol. 1990) measured isocitrate oxidation and corresponding respiration in isolated potato mitochondria, which proves that isocitrate is taken up by mitochondria. Isotopic dilution experiment with unlabeled isocitrate would be very valuable to test metabolic channeling.

There is little evidence for uptake of isocitrate at an appreciable rate by isolated potato mitochondria. Indeed the data in the Rasmussen and Moller paper referred to by the reviewer shows a rather low rate of oxidation of exogenous isocitrate (around an order of magnitude lower than the typical rate of oxidation of succinate and malate by potato mitochondria, for example). Given this uncertainty, we decided not to include isocitrate among the unlabeled organic acids tested.

The interactome study was based on proteins from Arabidopsis whereas the isotope dilution experiments were conducted in isolated potato mitochondria. Please discuss: i) how similar/different the protein sequences are between the two species;

The comparison of amino acid sequences of Arabidopsis and potato protein showed high similarity between the TCA cycle enzyme subunits from two organisms. The results of a BLAST search are shown in Supplementary Table 5 and the results have been described on P6 L24-27.

ii) the limitations of this study (can we assume that isolated potato mitochondria behave like other plant mitochondria?);

The limitation has been stated in the conclusion (P11 L5-7). However, we believe that our results still represent the composition of plant TCA cycle metabolon considering its similarity between organisms. This point has also been included.

iii) the flux around the TCA cycle was found to be absent in developing Brassica embryos (Schwender et al., JBC 2006), how does this finding compare to the present study?

We are unclear what the reviewer's point is here. Non-cyclic modes of the TCA cycle are well-known and the exact configuration of carboxylic acid metabolism is highly context-dependent (see Sweetlove, L. J., et al. (2010) Not just a circle: flux modes in the plant TCA cycle. *Trends Plant Sci* **15**, 462–470). In Brassica embryos cyclic flux is low and not absent. The dominant flux, however, is not cyclic but rather a linear pathway between OAA/malate and citrate which is exported to the cytosol to meet the high requirement for cytosolic acetyl CoA for elongation of lipids. This configuration is driven by the fact that Brassica embryos are oil-storing tissues. In heterotrophic starch-storing tissues such as potato tubers, a conventional cyclic flux is dominant. The effect of environment and tissue type / developmental stage on TCA cycle function and regulation is considered in the 'Discussion' section. We have now slightly expanded this section of the Discussion (P10 L20).

G. References: appropriate credit to previous work?

78 references are currently listed, and they adequately credit previous work.

H. Clarity and context: lucidity of abstract/summary, appropriateness of abstract, introduction and conclusions

Overall, the manuscript is well written. The abstract has a concise description of the study and the main findings. The introduction clearly presents the state of the knowledge about metabolons, evidence of metabolons, metabolons in TCA cycle demonstrated in other organisms...

Reviewer #4 (Remarks to the Author):

This manuscript by Zhang et al is an interesting and unique investigation carried out by employing multiple techniques to find out the interactome of TCA cycle in Arabidopsis.

While going through the manuscript I felt the need to see the mass Spectrometry data of the study. I encourage authors to submit the MS data as suppl. file. Without this data it is impossible

to check the quality of the Affinity purification-Mass Spectrometry (AP-MS) especially when the authors have used MS after ONLY one step of purification. My experience is that two-step purifications are better for MS.

We employed a single step purification in order to capture weak interactions and the weak specificity should be compensated by the combination of multiple techniques. This explanation has been added on P8 L3-8. The mass spectrometry data from the AP-MS assays have been shown as Supplementary Dataset 2.

Moreover, all the techniques of this manuscript such as cell culture, Y2H, split-LUC - all of them can be considered as in-vitro, at least technically speaking none is in-planta approach. Authors should give a justification why they preferred cell culture over transgenic Arabidopsis plants for the AP-MS.

The protein-protein interactions detected in three of four systems take place in mitochondria of living Arabidopsis cells. In this context the techniques can be considered *in vivo* rather than *in vitro*. And they are certainly more advanced than much of the work on the mammalian TCA cycle metabolon which is largely dependent on the aggregation of purified enzymes in a test tube. This has been discussed on P8 L32-34.

We agree that the intracellular conditions in the cell culture will be different from those in plant tissues. However, the cell culture system employed here generally shows much higher success rate to accumulate enough amounts of transgenic proteins than Arabidopsis plants. Additionally the system allows us to complete the selection process within a month, which would take at least more than half a year for Arabidopsis plants. These features are necessary for our high-throughput approaches. These points have been mentioned on P8 L3-8.

In the current study we focused on identifying possible protein-protein interactions between TCA cycle enzyme subunits. This has been clarified at the end of Introduction (P3 L41-42). The use of cell culture and "*in vitro*" approaches meets this aim. We are currently trying to establish a system to test protein-protein interactions in planta under different conditions. The results should be published in a separate paper.

Suppl. table legends need more information on what the number in different columns mean? We are sorry for insufficient description in the legends for supplemental materials. We have checked and revised them.

Check the manuscript for typos. One example is Table S6 - "Doner"

We are sorry for the typos. We have tried to spot typos throughout the manuscript including the legends of Supplementary materials.

REVIEWERS' COMMENTS:

Reviewer #1 (Remarks to the Author):

Although I still think the discussion should be more quantitative, the authors have adequately addressed my suggested edits.

Reviewer #2 (Remarks to the Author):

The authors give convincing answers to my comments and substantially improved the manuscript. I consider that this revised version can be accepted.

Reviewer #4 (Remarks to the Author):

The authors have addressed all the issues for this manuscript. Excellent job!

Response to referees

Essentially no issue is raised by referees. Regarding the comment on quantitative discussion by Reviewer #1, we are keen to establish quantitative relationship between compromise score and stability of protein-protein interaction. This would be tested published as a separate paper in a future as stated in the response to the reviewers' comment in the previous round.

REVIEWERS' COMMENTS:

Reviewer #1 (Remarks to the Author):

Although I still think the discussion should be more quantitative, the authors have adequately addressed my suggested edits.

Reviewer #2 (Remarks to the Author):

The authors give convincing answers to my comments and substantially improved the manuscript. I consider that this revised version can be accepted.

Reviewer #4 (Remarks to the Author):

The authors have addressed all the issues for this manuscript. Excellent job!